# Properties and Biodegradability of Films Based on Cellulose and Cellulose Nanocrystals from Corn Cob in Mixture with Chitosan

**DOI:** 10.3390/ijms231810560

**Published:** 2022-09-12

**Authors:** Monserrat Escamilla-García, Mónica Citlali García-García, Jorge Gracida, Hilda María Hernández-Hernández, José Ángel Granados-Arvizu, Próspero Di Pierro, Carlos Regalado-González

**Affiliations:** 1Faculty of Chemistry, Autonomous University of Querétaro, Cerro de las Campanas S/N, Las Campanas, Santiago de Querétaro 76010, Mexico; 2CONACYT—Center for Research and Assistance in Technology and Design of the Jalisco State, A.C. (CIATEJ), Av. Normalistas 800, Colinas de la Normal, Guadalajara 44270, Mexico; 3Department of Agricultural Sciences, University of Naples “Federico II”, Via Università, 100, 80055 Naples, Italy

**Keywords:** edible films, nanocellulose, biodegradability

## Abstract

The increase in consumer demand for more sustainable packaging materials represents an opportunity for biopolymers utilization as an alternative to reduce the environmental impact of plastics. Cellulose (C) and chitosan (CH) are attractive biopolymers for film production due to their high abundance, biodegradability and low toxicity. The objective of this work was to incorporate cellulose nanocrystals (NC) and C extracted from corn cobs in films added with chitosan and to evaluate their properties and biodegradability. The physicochemical (water vapor barrier, moisture content, water solubility and color) and mechanical properties of the films were evaluated. Component interactions using Fourier-transform infrared (FTIR) spectroscopy, surface topography by means of atomic force microscopy (AFM), biodegradability utilizing a fungal mixture and compostability by burying film discs in compost were also determined. The C-NC-CH compared to C-CH films presented a lower moisture content (17.19 ± 1.11% and 20.07 ± 1.01%; *w*/*w*, respectively) and water vapor permeability (g m^−1^ s^−1^ Pa^−1^ × 10^−12^: 1.05 ± 0.15 and 1.57 ± 0.10; *w*/*w*, respectively) associated with the NC addition. Significantly high roughness (Rq = 4.90 ± 0.98 nm) was observed in films added to NC, suggesting a decreased homogeneity. The biodegradability test showed larger fungal growth on C-CH films than on CH films (>60% and <10%, respectively) due to the antifungal properties of CH. C extracted from corn cobs resulted in a good option as an alternative packaging material, while the use of NC improved the luminosity and water barrier properties of C-CH films, promoting strong interactions due to hydrogen bonds.

## 1. Introduction

The world plastics production was estimated as 359 million tons in 2018, packaging materials as one of their main applications. The high plastic accumulation in the environment and its elevated biodegradation resistance has raised contamination concerns, encouraging the increase of the consumer demand for more sustainable materials. Therefore, alternatives have been sought to produce biodegradable, renewable and environmentally friendly materials at a competitive cost [1,2]. The development and application of biopolymer-based films from agricultural byproducts or waste food has increased lately due to concerns about the overexploitation of limited natural resources such as fossil fuels and the high environmental impact of packaging made from nonbiodegradable materials.

Biopolymeric films meet the general characteristics of quality and appearance for food products but also for public health, which increases consumer interest [1]. Polysaccharides have emerged as one of the main biodegradable polymers for film production due to their high abundance and low toxicity; additionally, they are generally easy to make and exhibit good mechanical properties [3]. Furthermore, biopolymers such as starch, cellulose, chitosan, pectin, casein, collagen and soy, among others, have been studied for their use as alternative to plastics, since some of them show biodegradation rates of months or even days [4]. 

Agro-industrial waste can be obtained from trees or plants such as sugar cane bagasse, corn and bamboo, among others. The main fibrous residues produced by agricultural activity are known as cellulose fibers, and they have a high potential to be used as reinforcement in the manufacture of packaging materials due to their abundance, low weight, biodegradability and low cost. Assuming that 40% of agricultural production comprises byproducts and that at least 10% of them can be obtained as fibers, millions of metric tons of fiber would be available every year [5].

Polysaccharides are characterized by forming colorless, tasteless, odorless and nontoxic films with good gas barrier properties and a high permeability to water vapor but a low diffusion of aromatic compounds and fatty substances [6]. Cellulose © is a linear homopolymer composed of β-D-1, 4-glucose units linked by glycosidic bonds. Cellulose comprises a reducing end of D-glucose, showing a C1-OH group with an aldehyde structure, while the nonreducing end has the terminal group C4-OH. The molecular structure is responsible for its chirality, hydrophilicity and degradability [7]. 

The addition of cellulose nanocrystals (NC) into other polymers may improve the intrinsic properties of these materials, such as mechanical, optical and barrier properties, among others [8]. NC are usually obtained by the sulfuric acid hydrolysis of cellulose and possess a rigid fibrillar structure [4]; even at low concentrations, they can be used as the reinforcing material of a polymeric matrix due to their nanometric size (3–50 nm) and the large exposed surface area [9]. NC are characterized by a large number of OH groups on the surface, which favors the formation of hydrogen bonds, facilitating the interaction of cellulose with other polymers. NC have been used to modify the barrier properties of packaging materials and to promote reduced friction, preserving the optical properties of the materials [10].

Global corn production achieved 1.162 Gtons in 2020 [11], whereas, according to FAO, the total production of corn in Mexico averaged 27.5 million tons in 2021 [10]. The corn cob is the part of the corn ear remaining after removing the corn grains, which accounts for 75–80% of the ear of corn [12]. Corn cobs are used mainly as fodder for ruminants, as support to reduce soil erosion, for xylitol and ethanol production and as a substrate for the production of xylanase. However, its use in these areas is limited, which has led to the burning of this byproduct or its spreading outdoors, generating an environmental pollution problem [13,14]. A corn cob is mainly composed of cellulose (40–44%), hemicellulose (31–33%) and lignin (16–18%) [15], and for this reason, it is considered a good source of cellulose.

Chitosan ((β1→4) linked residues of N-acetyl-2 amino-2-deoxy-D-glucose (glucosamine, GlcN) and 2-amino-2-deoxy-D-glucose (N-acetyl-glucosamine, GlcNAc) residues) is a polysaccharide obtained from the partial deacetylation of chitin, which exhibits antioxidant, antitumor and antimicrobial properties. Its charge density and solubility is dependent on the degree of deacetylation, molecular weight and solution pH [16]. Chitosan (CH) has the good ability to form films, displaying a low permeability to gases (CO_2_ and O_2_) and good mechanical properties but high permeability to water vapor. The objective of the present work was to determine the physicochemical, mechanical and barrier properties, as well as topography and biodegradability, of films using corn cobs as a source of cellulose and cellulose nanocrystals in a mixture with chitosan.

## 2. Results and Discussion

### 2.1. Yield of Cellulose and Cellulose Nanocrystals

The yield of C obtained from corn cobs in this work was lower (21.28% ± 1.31% by weight) than those reported by other authors of about 30% [14]. The low yield may be attributed to material losses due to repeated washings. The yield of NCs from the corn cob using acid hydrolysis and ultrasound was 5.70 ± 1.71 (*w*/*w*), which coincides with those reported by other authors that are close to 6% (*w*/*w*). The low NC recovery may be associated with the poor selectivity of ultrasound waves acting on both the amorphous and crystalline regions that may explain NC yields lower than 10% (*w*/*w*) [17].

### 2.2. Films Characterization

#### 2.2.1. Physical Properties

The moisture contents (MC) of the three different films produced presented significant differences (*p* < 0.05) (Table 1). From these results, it is observed that the highest MC was shown by the CH film (33.35 ± 1.20%, *w*/*w*), while the incorporation of C and NC into the films’ formulation reduced this parameter. MC depends on the hydrophilic nature of the films; the larger the number of exposed hydroxyl groups, the greater affinity of water molecules towards the material, with a consequent increase in the MC [18].

The MC values reported here are similar to those reported for polysaccharide matrices (≈20% *w*/*w*) [19]. The addition of C and NC to the CH resulted in films with reduced MC, which may be due to interactions occurring through hydrogen bonds between the C and CH that extensively block the hydroxyl groups trying to interact with the surrounding water molecules. Moreover, it has been reported that the use of NC in the film-forming solutions results in water access restriction into the formed film matrix, due to its crystalline structure [20].

C-CH and C-NC-CH films’ water solubility (Ws) were not significantly different (*p* < 0.05). The reports on the Ws value of CH-NC films are in agreement with the values shown here (21% by weight) [21]. The Ws values of the C-CH and C-NC-CH films were significantly lower than that of the CH films probably due to chemical crosslinking within the hydrophilic polymeric matrix. From Table 1, it can be seen that the addition of either C or NC to CH restricts the diffusion of water molecules within the polymeric structure, leading to reduced Ws of the films. It is known that NC increases the cohesion of the film components, which, in turn, reduces the hydroxyl groups available to interact with water, and based on this effect, it has been proposed that Ws is inversely proportional to the NC concentration [22].

The C-NC-CH film exhibited the lowest water vapor permeability (WVP) (1.05 ± 0.15 × 10^−12^ g m^−1^ s^−1^ Pa^−1^), while the CH film showed the highest value (7.8 ± 0.20 × 10^−12^ g m^−1^ s^−1^ Pa^−1^) (Table 2). Films made from polysaccharides such as CH are generally quite hydrophilic, so they have poor water barrier properties [23]. The C-CH and C-NC-CH films presented WVP values similar to those reported by Cazón et al. [24] (6.6 × 10^−13^–1.6 × 10^−11^ g m^−1^ s^−1^ Pa^−1^), but a significantly higher value was reported for CH (3.65 × 10^−10^ g m^−1^ s^−1^ Pa^−1^) [25]. The films containing NC showed better water barrier properties than those based on C-CH. This may be associated with interactions between the polymeric matrix and the NC, which reduced the availability of the hydrophilic groups able to interact with the water molecules, resulting in a reduced water vapor transfer rate [24].

#### 2.2.2. Mechanical Properties

The CH film showed the lowest thickness (474.90 ± 46.27 μm), which, upon the addition of C, increased significantly (Table 2). However, the C-CH and C-NC-CH films did not significantly exhibit different thicknesses (*p <* 0.05).

The tensile strength (TS) of the films did not show significant differences (*p* < 0.05) (Table 2). The films made from CH only exhibited the highest TS value (1.093 ± 0.250 MPa), followed by the C-NC-CH film. The effect of the C addition to CH films decreased TS to a higher extent than the NC addition, suggesting that NC acts by promoting strong interactions and hydrogen bond formation with the polymeric matrix. In addition, glycerol helps to increase the dispersion and material component interactions [26]. C-NC-CH film’s TS was relatively low, which was attributed to agglomerate formations when NC was added because of induced stress points in the polymeric matrix [27].

#### 2.2.3. Color Parameters

The parameters evaluated corresponded to the CIELAB color space: a* (coordinates red to green), b* (coordinates yellow to blue) and L* (luminosity). After NC incorporation, the a* and b* values of the films increased (Table 3), indicating that the C-NC-CH film is more reddish and yellowish, whereas the CH films showed the lowest b* value because of greater blue color, in agreement with other reports [19].

A high lightness (L*) value indicates clearer and more transparent films. The C-CH films exhibited significantly low L* values, associated with the CH polymer matrix bond formation with other film compounds (Table 4).

These results indicate that the CH film tends to be more transparent (Figure 1) [28]. Film colors may be described using other parameters, such as ΔE, which indicates the extent of the total color difference relative to the white reference plate. The ΔE of the films with C-CH and C-NC-CH was significantly higher than that of CH, corroborating that the CH film was the most transparent [25].

#### 2.2.4. Films Topography

The CH film (Figure 2C) presented the smoothest surface (Ra = 0.95 nm ± 0.09; Rq = 1.2 ± 0.07 nm), although not significantly different from the C-CH film (*p* < 0.05) (Figure 1A), whereas the NC-containing film (Figure 2B) was the roughest (Ra = 3.97 ± 0.85 nm; Rq = 4.90 ± 0.98 nm). Upon the addition of NC, aggregates were formed, which explains the rough surfaces of C-NC-CH films, decreasing their homogeneity (Figure 2) [29].

Figure 3 shows micrographs of NC and CH-NC films. Figure 3A shows the particle size of NC obtained by AFM, where 15–20 particles were selected from different areas of the film, ranging between 74.63 and 128.85 nm.

According to Boukouvala et al. [30], cellulose may be considered a nanocrystal if the crystalline particles range from 1 to 1000 nm. The observed particles size is the result of the type of extraction applied, and some authors consider that the size is proportional to the degree of polymerization. Hydrolysis carried out with sulfuric acid (64% *v*/*v*) allowed crystal formation [31].

Figure 3B shows the micrograph of a NC-CH film for an area of 5 × 5 mm, in which the formation of agglomerates can be observed. According to Börjesson et al. [31], for films made with NC and dried by evaporation in contact with the air, there is the induction of NC agglomerates during film formation. 

#### 2.2.5. Films Components Interactions Evaluated by FTIR

The films’ FTIR spectrograms are shown in Figure 4. The spectrograms of the films produced from corn cob cellulose (C), cellulose–chitosan (C-CH), and cellulose–cellulose nanocrystals–chitosan (C-NC-CH) are shown in Figure 4A–C, respectively. The corn cob cellulose spectrogram shows the O-H tension vibration characteristic peak at 3313 cm^−1^, a band due to –C-H tension vibration is located at 2933 cm^−1^ and one band across the O-H bond of adsorbed water is located at 1639 cm^−1^, while the band corresponding to –CH_2_ snipping is at 1422 cm^−1^. A fluttering –CH_2_ band is observed at 1319 cm^−1^, while the pyranose ring strain vibration band C-O-C is located at 1031 cm^−1^ [32].

CH-C (Figure 4B) and C-NC-CH (Figure 4C) films show characteristic CH peaks at 1648, 1544 and 1411 cm^−1^ that correspond to the stretching of C=O (amide I), to N–H bending (amide II) and to HN-CO stretching (amide III), respectively. The absorption peak at 1030 cm^−1^ is due to C–O stretching; the peaks between 2920 and 2850 cm^−1^ are related to the amino group, and the peaks in the range of 3600–3200 cm^−1^ correspond to the O–H and N–H stretch bands. It is well-known that characteristic NC absorption bands appear at 3000–2800 cm^−1^ (C-H stretching of the CH_2_ and CH_3_ groups) and at 3455–3230 cm^−1^ (O-H stretching) [25,33].

All spectra showed peaks in the 850–640 cm^−1^ region that are assumed to be characteristic of the corn cob composition. When chitosan was added, the characteristic peaks of the amino group appeared at 2920 cm^−1^ and those of the amides in the region of 1650–1410 cm^−1^, whereas the cellulose bands at 2933 and 1319 cm^−1^ disappeared (Figure 4A,B) [25].

The broad peak in the region between 3500 and 3300 cm^−1^ is attributed to the O–H hydrogen bond stretching vibration, indicating strong interactions between NC and CH through hydrogen bonds. The bands in the region from 1650 to 1410 cm^−1^ exhibited a higher intensity in the C-NC-CH film. In addition, the 1030 cm^−1^ peak also presented a higher intensity with the NC addition (Figure 4B,C). These results indicate that the link between chitosan and NC occurs through hydrogen bonds [21,29].

#### 2.2.6. Films Biodegradability

Biopolymer biodegradation, such as cellulose and chitosan, involves the hydrolytic or enzymatic breakage of their backbone structure, including the removal of hydrogen bonds, with the formation of CO_2_, CH_4_, water, biomass and other natural substances as the products [34].

Figure 5 shows that, following the 28 days of the biodegradability test, there was a large fungal growth in the positive control (Figure 5D) and in the C-CH film (Figure 5A) that was classified as 4, indicating complete coverage. The film with NC (Figure 5B) was classified as 3, because it displayed about 50% of its surface covered by fungi, whereas the CH film exhibited the least fungal growth and was classified as 1 (Figure 5C).

Based on these results, it was shown that the C-CH and C-NC-CH films and, to a lesser extent. the CH film, acted as a substrate for the fungi mixture used, which indicated that the three formulations may be classified as biodegradable materials. The delay in fungal growth on the surfaces of the films was attributed to the presence of CH, which possesses antifungal properties. CH acts, promoting the permeabilization of the fungi cell wall through electrostatic interactions between the positive charges of the protonated amino groups and the negative charge of the fungal cell wall. Permeabilization triggers the loss of intracellular material, leading to cell death and, thus, the inhibition of fungal growth [35,36]. 

From Figure 5, it can be seen that the C-NC-CH film (Figure 5B) showed biodegradability to a lower extent than the C-CH film (Figure 5A), attributed to the more hydrolytic resistance of the crystalline regions of NC [27].

The progress of biodegradability can be observed in Figure 6 at 5 d, 10 d, 15 d, 20 d, 25 d and 28 d after the inoculation. According to the standard, testing can be completed in less than 28 d for samples showing a growth index of two or more (10–30% of the area covered), and thus, the study could be stopped after 10 d for the cellulose film and the control (filter paper). However, no growth was observed on the surface of the chitosan film after 28 d of analysis, while the NC film presented >10% of its surface covered by fungi at 20 d.

#### 2.2.7. Films Compostability

When the useful life of bioplastics ends, one of the most widely used forms of disposal is through composting. It can be observed from Table 4 that the films display a lower percentage of compostability than the filter paper (100%). Natural polymers such as cellulose and chitosan are often assumed to be biodegradable and environment friendly. However, biodegradable materials are not necessarily compostable, since the latter requires specific settings to break down, disintegrating into small fragments, and their biodegradation products do not represent damage to the environment in terms of ecotoxicity [37,38].

The films produced in this work did not show significant differences in their compostability (37.35–43.75% by weight). The difference in the compostability extent of the films relative to the filter paper (FP) used as the control is attributed to the cellulose content. The composts comprise a high microbial population that produces enzymes that degrade complex molecules such as cellulose [39]. It has been reported that, under composting conditions, CH takes approximately 70 d for 100% degradation, which is longer than the time taken by cellulose; the compostability values of the films were lower than that shown by filter paper [40]. 

Other studies have reported that films with NC (30% *p*/*p*) exhibit less than 5% compostability after 15 d, which is significantly lower than the value obtained in this work, associated with the low NC concentration incorporated into the C-NC-CH films [41]. In the case of CH films, there is a compostability report of only 15% after 12 d, and this difference may be attributed to the compostability conditions, which vary from one study to another [42].

## 3. Materials and Methods

### 3.1. Materials

Corn cobs (*Zea mays*, spp. mays) were supplied by the community of Texcatepec in the municipality of Chilcuautla (Hidalgo, México). Chitosan of medium molecular weight showing ≥90% deacetylation was acquired from Chemsavers (Luefield, VA, USA). All other chemicals were of analytical grade and commercially supplied.

### 3.2. Methods

#### 3.2.1. Cellulose Extraction

Cellulose extraction was carried out in corn cobs according to Melikoğlu et al. [43], with some modifications. The corn cobs were dried in an oven (Binder, WTB DB 115, Tuttlingen, Germany) at 50 °C for 24 h; then, two grinding processes were used, the first involving a hammer mill using a 3-mm mesh size (Model Qvn, México), and the second was a coffee grinder (Krups, Mod. GX4100, Solingen, Germany). The final particle size was about 2.83 mm, obtained by sieving the powder using a No. 7 mesh (Tyler Standard, OH, USA).

Ground corn cob (3.3 g) was placed in 100 mL of 10% (*w*/*v*) NaOH and heated at 55 °C for 3 h with continuous stirring using a magnetic stirrer (Barnstead Thermolyne, Dubuque, IA, USA). Subsequently, the insoluble residue was filtered and washed with distilled water until neutral pH was achieved and dried in the oven (Binder) at 60 °C for 24 h; after completing this process, most lignin was removed. Subsequently, the dry sample was placed in 1% (*v*/*v*) NaClO and heated at 95 °C for 1 h with constant stirring, repeating this process twice. The sample was filtered and washed with distilled water until a neutral pH and dried at 60 °C (Binder oven) for 24 h. 

#### 3.2.2. Cellulose Nanocrystals Production

The NC were obtained by acid hydrolysis following Zhang et al. [44], with some modifications. Three grams of the extracted cellulose were placed in 60 mL of 64% (*v*/*v*) H_2_SO_4_ at 45 °C for 1.5 h with constant stirring. The hydrolysis was stopped by adding ice-cold distilled water in a 1:10 ratio (suspension:H_2_O, *v*/*v*), and the mixture was stirred without heating for 10 min. Then, it was centrifuged (Eppendorf, 5810 R, Hamburg, Germany) at 4 °C and 4000× *g* for 5 min, repeating this process four times to remove the acid. The resulting insoluble residue was diluted with distilled water and dialyzed at room temperature for 72 h using a 12-kDa cut-off dialysis bag (Sigma-Aldrich, St. Louis, MO, USA). The nanocrystal suspension was sonicated (Branson, Mod. 5510, Danbury, CT, USA) at 25 °C for 10 min. The resulting solution was homogenized using an Ultra-Turrax (IKA, Mod. T25 Basic S1, Staufen, Germany) at 9500 rpm for 2 min. Finally, 1% (*v*/*v*) NaClO solution was added and kept under constant stirring for 1 h, centrifuged (4 °C, 4000× *g*, 5 min), repeated 4 times to obtain a neutral pH and, finally, the pellet was dried 60 °C (Binder oven) for 24 h.

#### 3.2.3. Cellulose (C) and Cellulose Nanocrystals (NC) Yield

The cellulose yield was obtained following the methodology of Gupta et al. [45], with some modifications. An acid hydrolysis was performed by placing the extracted cellulose in 5% (*v*/*v*) sulfuric acid at 20 g L^−1^ and heated at 100 °C for 3 h, followed by quantification of the glucose concentration using a glucose kit (GAGO20, Merck, Darmstadt, Germany). The glucose yield was obtained according to Equation (1).
(1)$$Y_{C}\left(\%\right)=\frac{C_{G}*V}{S}*100$$
where *Y_C_* is the cellulose yield, *C_G_* is the glucose concentration (g L^−1^), *V* is the volume (L) and *S* is the added substrate (g). The NC were dried at 105 °C until a constant weight was reached. Subsequently, the *NC* yield was obtained from Equation (2).
(2)$$Y_{NC}\left(\%\right)=\frac{P_{NC}}{P_{C}}*100$$
where *Y_NC_* is the *NC* yield in relation to the extracted cellulose, *P_C_* is the weight of the initial cellulose and *P_NC_* is the mass of nanocrystals obtained [46].

#### 3.2.4. Nanocellulose Crystals Size

Crystal sizes were determined using atomic force microscopy (AFM: Park NX10, Seoul, Korea), applying the no contact method and using an aluminum-coated silicone tip PPP-FMR (Nanosensors, PointProbe, Neuchatel, Switzerland) with a resonance frequency of 286–362 kHz and a spring constant of 20–80 N m^−1^. Samples of 0.5 × 0.5 cm were analyzed, and three 5 × 5 μm areas were scanned at a speed of 1 Hz with a resolution of 256 × 256 pixels [47]. A particle size of 20 particles was determined for each area of 5 μm × 5 μm, and the measurements were made on 5 different films and at 3 different areas of each one.

#### 3.2.5. Edible Films Production

Three different solutions were prepared: the first was a 1% (*w*/*v*) solution of chitosan (CH) in 0.5% (*v*/*v*) acetic acid, followed by heating at 90 °C for 1 h under constant stirring [48]. Subsequently, the cellulose and NC solutions were prepared according to Ghosh et al. [49], with some modifications. The second solution was prepared by dissolving 1.5% (*v*/*v*) of cellulose (C) in 0.5% (*v*/*v*) acetic acid with constant stirring for 1 h. Finally, 0.3% (*w*/*v*) NC and 1.2% (*w*/*v*) C were dissolved in 0.5% (*v*/*v*) acetic acid under constant stirring for 1 h. From these solutions, three different films were made by the casting method. The first film was made from the CH solution, and another film contained the mixture of C and CH in a 1:1 ratio (*v*/*v*), while the third film was produced from the mixture of C-NC:CH (1:1 ratio, *v*/*v*). To each filmogenic solution, 1% glycerol was added as a plasticizer, stirred for 90 min and then homogenized using the Ultra-Turrax, at 9500 rpm for 2 min, followed by drying at 25 °C for 48 h.

#### 3.2.6. Films Characterization

##### Moisture Content

The moisture content (MC) of the different films was determined according to Gutiérrez [37]. The dry weight (W_D_) of the films was obtained by cutting 2 × 2 cm squares of each film, followed by heating at 105 °C for 24 h and weighing each piece using an analytical balance (Sartorius, BA 110 S, Bohemia, NY, USA); the samples were initially weighed to obtain the wet weight (W_W_). The MC was calculated using Equation (3).
(3)$$\mathrm{MC}=\frac{W_{w}-W_{D}}{W_{W}}$$

##### Water Solubility

Square pieces of the films were cut (20 mm × 20 mm) and dried at 105 °C to a constant weight (W_0_). The dried films were immersed in 50 mL deionized water for 15 h. After this time, the solution was filtered using previously dried and weighed filter paper, and the undissolved films were dried at 105 °C for 24 h until constant weight (W_1_). The water solubility (W_S_) was calculated using Equation (4) [50].
(4)$$W_{S}\;\left(\%\right)=\frac{W_{0}-W_{1}}{W_{0}}*100$$

##### Film Thickness

The films’ thickness were measured using a micrometer (Mitutoyo, 293–344-30, Aurora, IL, USA) at ten different random positions along the surfaces of the films. Values are reported as the mean ± standard deviation of five replicates.

##### Water Vapor Permeability

Water vapor permeability (WVP) was evaluated according to Escamilla-García et al. [51]. Permeability cells with known cross-sectional areas (A) were used, in which the different films with known thicknesses (L) were fitted, then the cells were placed inside a desiccator at constant temperature. To generate a water vapor pressure difference (ΔP), different saturated solutions were poured inside the cells (NaCl; RH = 75%) and inside the desiccator (KNO_3_; RH = 95.6%). Weight variations (ΔW) were recorded every 15 min until the cell reached a constant weight, and the time (t) was recorded. WVP was obtained by using Equation (5).
(5)$$\mathrm{WVP}=\frac{\Delta W}{t\;A}*\frac{L}{\Delta P}\;$$

##### Mechanical Properties

Tensile strength at the breaking point (TS) was determined using a texturometer (Brookfield, CT3, Middleborough, MA, USA). The films were cut into 80 × 25 mm rectangles, which were placed between two clamps, with an initial grip gap of 97.9 mm. During the measurements, an activation load of 4 N was established, and the films were stretched at a speed of 0.3 mm/s. The results were processed using TexturePro CT 1.6 software (Brookfield, Middleborough, MA, USA). The TS was calculated using Equation (6).
(6)$$\mathrm{TS}\;(\mathrm{MPa})=\frac{L}{A}\;$$
where L is maximum load (N), and A is the cross-sectional area of the film (m^2^).

##### Color

Films’ colors were evaluated following Escamilla-García et al. [47] using a colorimeter (Konica Minolta, CR-400, Ramsey, NJ, USA) with a D65 light source at a viewing angle of 10° standardized using a white reference plate (L* = 90.9, a* = 0.021 and b* = 0.0376). Films placed on this plate were measured at five different positions along the surfaces of the films (center and outer parts), avoiding the edges. Color differences (ΔE), measured as the magnitude of the vector resulting from the three components: luminosity difference (ΔL), red–green chromaticity difference (Δa) and yellow–blue chromaticity difference (Δb) were calculated using Equation (7).
(7)$$\;\Delta E=\sqrt{{(\Delta a)}^{2}+(\Delta {b)}^{2}+(\Delta {L)}^{2}}\;$$
where Δa = a_i_ − a, Δb = b_i_ − b and ΔL = L_i_ − L. The subscript i is the reference value of each parameter.

##### Films’ Topography

The films’ topographical characteristics were determined using an atomic force microscope (Park NX10, Korea), applying the no contact method and using an aluminum-coated silicone tip PPP-FMR (Nanosensors, Switzerland) with a resonance frequency of 286–362 kHz and a spring constant of 20–80 N m^−1^. Samples of 0.5 × 0.5 cm were analyzed, and three 1 × 1 μm areas were scanned at a speed of 1 Hz with a resolution of 256 × 256 pixels [47]. The image analysis and the roughness parameters Ra and Rq were obtained using the Smartscan program (Czech Metrology Institute, CZE).

##### Fourier Transform Infrared Spectroscopy (FTIR)

FTIR spectra were obtained using an IR2 Module spectrophotometer (Horiba Jobin Ybon, Kyoto, Japan) equipped with a diamond ATR objective at a resolution of 4 cm^−1^, a range of 400–4000 cm^−1^ and taking 32 scans per reading. The spectra were analyzed using the Spectragryph 1.1 program (Spectroscopy Ninja, USA).

##### Films’ Biodegradability

The films’ biodegradability were evaluated following the ASTM G21-09 method [52]. For this test, *Aspergillus niger*, *Penicillium pinophilum*, *Chaetomium globosum*, *Gliocladium virens* and *Aureobasidium pullulans* were used. Fungi were inoculated in petri dishes in a salt-enriched agar (15 g L^−1^ agar, 0.7 g L^−1^ K_2_HPO_4_, 0.7 g L^−1^ KH_2_PO_4_, 0.7 g L^−1^ MgSO_4_·7H_2_O, 1.0 g L^−1^ NH_4_NO_3_, 0.005 g L^−1^ NaCl, 0.002 g L^−1^ FeSO_4_·7H_2_O, 0.002 g L^−1^ ZnSO_4_·7H_2_O and 0.001 g L^−1^ MnSO_4_·H_2_O) and were incubated at 30 °C for 28 d. Once the fungi sporulated, individual spore suspensions were prepared by pouring 10 mL of 0.05 g L^−1^ sterile solution of Tween 80 on each subculture, scraping its surface and emptying the spore load into 45 mL of sterile water. The spores were washed three times and diluted with a salt-enriched sterile solution to obtain a suspension of 1.0 × 10^6^ ± 0.2 × 10^6^ spores mL^−1^, and equal volumes of each suspension were used to obtain a mixture of spores.

Films squares (2.5 cm × 2.5 cm) were placed in Petri dishes containing salt-enriched agar, and 1 mL of spore mixture was inoculated; then, the plates were incubated at 30 °C and 85% RH for 28 d. During this time period, a visual inspection of fungi growth was carried out, and on day 28, a rating was given according to Table 5. Filter paper squares were used as a positive control. Additionally, a control plate without spores mixture was included.

##### Films’ Compostability

The films’ compostability was evaluated using the method described by Gutiérrez [37], with some modifications. Disks of 0.6 cm in diameter from each film were cut, and the initial dry matter content of each disk (Wi) was determined by oven drying at 105 °C until a constant weight. The compost was prepared according to Sintim et al. [53], consisting of a carbon–nitrogen ratio of 25–30:1 and a moisture content between 55 and 65% (*w*/*w*). The base compost used was commercially acquired (Compost-on, CDMX, Mexico). The mixture contained (*w*/*w*) broiler litter (28%), yard wastes (28%), manure (28%), animal bedding (14%) and fish carcasses (2%). The compost (500 g) was placed in plastic containers of 11.5 cm in diameter and 7.6 cm in height. At an approximate depth of 4.5 cm, the film disks and filter paper were placed, and four film disks were buried in each container: C-CH, C-NC-CH and CH, placing a marker on the surface of the compost to indicate the position of the film disks. On days 4, 8 and 12, after the initial time (day 0), the final dry matter content of each disk was obtained (Wf). The tests were carried out at room temperature (25 °C ± 2 °C) and 60–70% relative humidity. The compostability percentage (CP) was calculated using Equation (8).
(8)$$\mathrm{CP}\;(\%)=\frac{\mathrm{Wi}\;-\;\mathrm{Wf}}{\mathrm{Wi}}\;\times \;100$$

##### Statistical Analysis

To obtain representative results, five different sections of three different films were tested. The results are the averages of these measurements ± standard deviation. Data were subjected to one-factor analysis of variance and analyzed by the comparison of means using Tukey’s test (*p* < 0.05) using the SigmaPlot 14.0 program (Systat, Chicago, IL, USA).

## 4. Conclusions

The extraction of C and NC from corn cobs using alkaline conditions, bleaching and acid hydrolysis treatments produced low yields. Atomic force microscopy detected that films containing NC were the least homogeneous, associated with aggregate formations. Compared with the CH films, the C-CH and C-NC-CH films showed lower moisture contents, water solubility and luminosity, larger thicknesses and roughness values and higher biodegradability. The addition of NC conferred to the films better water barrier properties and increased the luminosity of the C-CH films, while the tensile strength and compostability were not significantly different from the other films. The biodegradability of the CH films was much lower than that shown by the C-CH and C-NC-CH films. From the FTIR spectroscopy, it was observed that adding CH to C films produced decreased tension vibrations of the OH groups, while strong interactions due to hydrogen bonds were revealed by the NC-CH films.

## Figures and Tables

**Figure 1 ijms-23-10560-f001:**
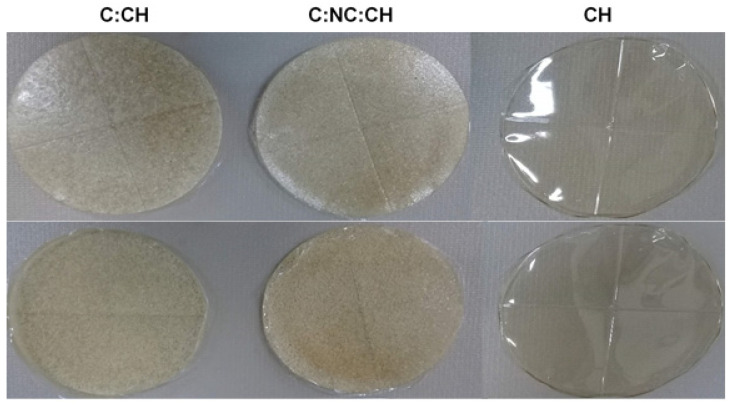
Appearance of edible films. C: Cellulose, NC: Cellulose Nanocrystals and CH: Chitosan.

**Figure 2 ijms-23-10560-f002:**
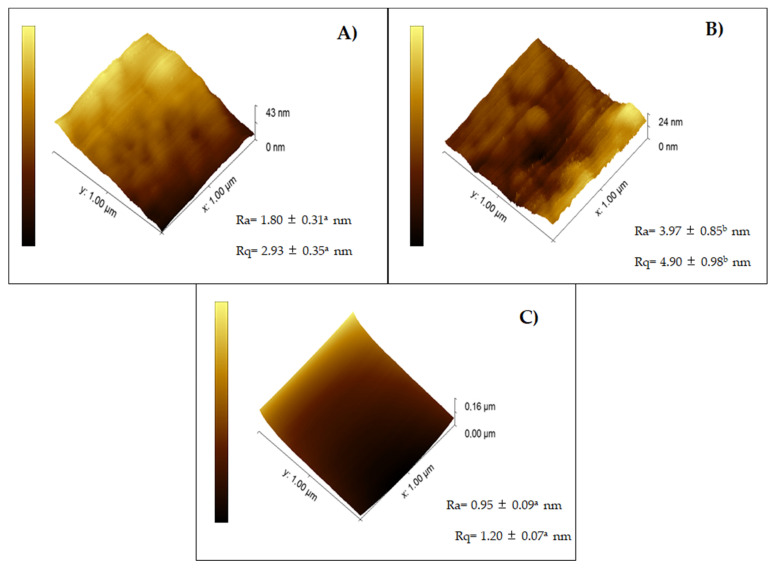
Films’ topography. (**A**) Cellulose–Chitosan film, (**B**) Cellulose–Nanocrystals–Chitosan film and (**C**) Chitosan film. Ra: Arithmetic roughness average of the surface; Rq: Root mean square average of the profile heights over the evaluation length. Different lowercase letters (a and b) next to the reported values in the same parameter indicate significant differences (*p* < 0.05).

**Figure 3 ijms-23-10560-f003:**
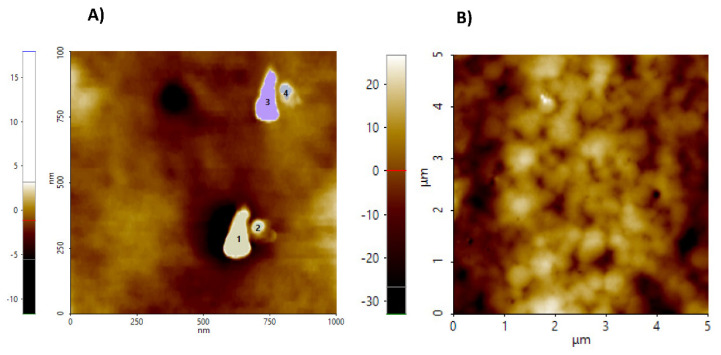
**(A**) Cellulose nanocrystals showing the selected particles size range of 74.63–128.85 nm. (**B**) Films of cellulose nanocrystals and chitosan (NC-CH).

**Figure 4 ijms-23-10560-f004:**
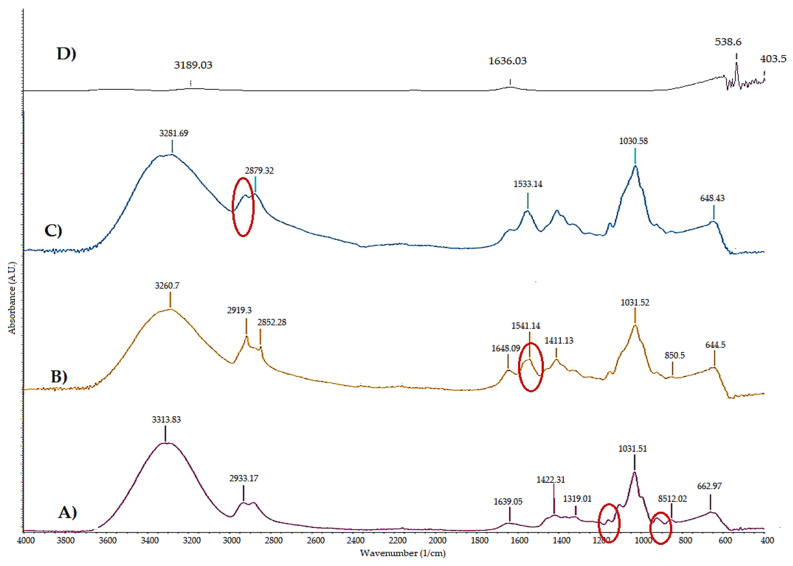
Films’ spectrogram. (**A**) Corn cob cellulose (C), (**B**) Cellulose–Chitosan (C-CH), (**C**) Cellulose–Cellulose nanocrystals–Chitosan (C-NC-CH) and (**D**) Chitosan (CH).

**Figure 5 ijms-23-10560-f005:**
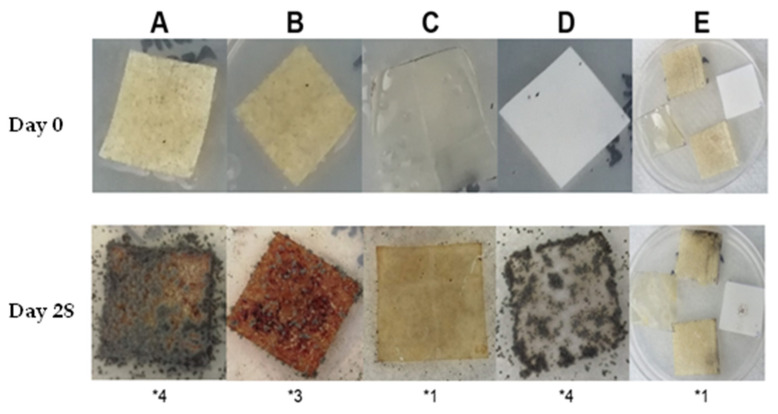
Films’ biodegradability observed on days 0 and 28 after spore mixture inoculation, comprising *Aspergillus niger*, *Penicillium pinophilum*, *Chaetomium globosum*, *Gliocladium virens* and *Aureobasidium pullulans*. (**A**) Cellulose–Chitosan, (**B**) Cellulose–Cellulose nanocrystals–Chitosan, (**C**) Chitosan, (**D**) Filter paper and (**E**) Negative control films were placed in a non-inoculated medium. The numbers below each image correspond to the growth scale described in Table 5.

**Figure 6 ijms-23-10560-f006:**
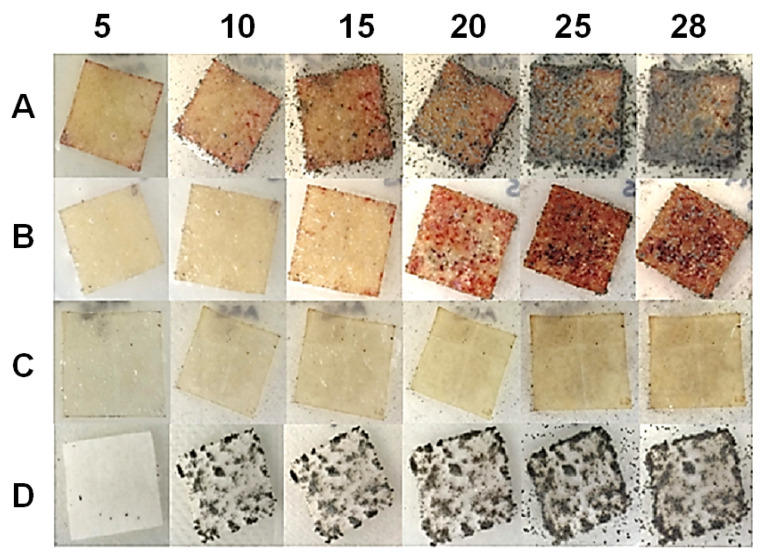
Biodegradability progress of cellulose films (**A**), cellulose nanocrystals (**B**), chitosan (**C**) and filter paper (**D**) observed on days 5, 10, 15, 20, 25 and 28 after inoculation of the spore mixture.

**Table 1 ijms-23-10560-t001:** Films’ physical properties.

Film	MC (% *w*/*w*)	Ws (% *w*/*w*)	WVP × 10^12^ (g m^−1^ s^−1^ Pa^−1^)
C-CH	20.07 ± 1.02 ^a^	17.44 ± 0.57 ^a^	1.57 ± 0.10 ^a^
C-NC-CH	17.19 ± 1.11 ^b^	19.80 ± 1.62 ^a^	1.05 ± 0.15 ^b^
CH	33.35 ± 1.20 ^c^	35.73 ± 6.27 ^b^	7.8 ± 0.2 ^c^

Results are reported as the mean ± standard deviation. Different lowercase letters (a–c) next to the reported values within columns indicate significant differences (*p* < 0.05). C: Cellulose, NC: Cellulose nanocrystals, CH: Chitosan, MC: Moisture content, Ws: Water solubility and WVP: Water vapor permeability.

**Table 2 ijms-23-10560-t002:** Films’ mechanical properties.

Films	Thickness (μm)	Tensile Strength (MPa)
C-CH	615.93 ± 30.03 ^a^	0.793 ± 0.228 ^a^
C-NC-CH	632.70 ± 15.4 ^a^	0.836 ± 0.129 ^a^
CH	474.90 ± 46.27 ^b^	1.093 ± 0.250 ^a^

Results are reported as the mean ± standard deviation. Different lowercase letters (a and b) next to the reported values within columns indicate significant differences (*p <* 0.05). C: Cellulose, NC: Cellulose nanocrystals and CH: Chitosan.

**Table 3 ijms-23-10560-t003:** Film colors.

Film	*a**	*b**	*L**	Δ*E*
C-CH	−1.88 ± 0.02 ^a^	19.09 ± 0.01 ^a^	87.67 ± 0.17 ^a^	19.42 ± 0.04 ^a^
C-NC-CH	−0.95 ± 0.02 ^b^	19.18 ± 0.02 ^b^	89.80 ± 0.66 ^b^	19.21 ± 0.04 ^b^
CH	−0.96 ± 0.00 ^b^	7.42 ± 0.01 ^c^	97.69 ± 0.04 ^c^	10.08 ± 0.03 ^c^

Results are reported as the mean ± standard deviation. Different lowercase letters (a–c) next to the reported values within columns indicate significant differences (*p <* 0.05). C: Cellulose, NC: Cellulose nanocrystals and CH: Chitosan.

**Table 4 ijms-23-10560-t004:** Films’ compostability.

Films	Compostability (%)
C-CH	37.35 ± 1.88 ^a^
C-NC-CH	43.75 ± 7.18 ^a^
CH	40.74 ± 19.36 ^a^
FP	100 ± 0.10 ^b^

Results are reported as the mean ± standard deviation. Different lowercase letters (a and b) next to the reported values within columns indicate significant differences (*p* < 0.05). C: Cellulose, NC: Cellulose nanocrystals, CH: Chitosan and FP: Filter paper.

**Table 5 ijms-23-10560-t005:** Fungal growth evaluation scale for the biodegradability test.

Growth Observation *	Scale
None	0
Trace growth (<10%)	1
Light growth (10–30%)	2
Medium growth (30–60%)	3
High growth (60%-fully covered)	4

* Corresponds to the extent of the film surface covered by fungi.

## Data Availability

Not applicable.

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
