# Peer review of "Properties and Biodegradability of Films Based on Cellulose and Cellulose Nanocrystals from Corn Cob in Mixture with Chitosan"

_ijms, 2022, doi:10.3390/ijms231810560_

Round 1
Reviewer 1 Report
The manuscript entitled Properties and biodegradability of films based on cellulose and cellulose nanocrystals from corn cob in mixture with chitosan present interesting results related to the characterization of films. The manuscript is well written and the results well presented. However, there are minor comments that authors must attend to prior to acceptance. Below are the comments.
-What is the rationale for showing table 1? The results must be presented in the main text.
- The quality of the images in figure 1 must be improved. Also, the figure must be depicted deeply and discussed.
-Line 193. Which is the depiction that belongs to A), B), and C)?
-Section 2.2.6. What was the rationale for monitoring the biodegradability only on day 0 and day 28? It could be interesting to evaluate the biodegradation of the films between days 0 and 28.
-Revise the order of the section numbers.
-Section 4.2.2. How do authors evidence they extracted nanocrystals?
Author Response
Q1. What is the rationale for showing table 1? The results must be presented in the main text.
Response
Comment acknowledged.
We removed table 1 and described the performance results in section 2.1 Line 104-111, as follows:
“The yield of C obtained from corn cob in this work was lower (21.28%±1.31%, by weight) than those reported by other authors of about 30% [14]. The low yield may be attributed to material losses due to repeated washings. The yield of NCs from the corn cob using acid hydrolysis and ultrasound was 5.70±1.71 (w/w), which coincides with those reported by other authors that are close to 6% (w/w). The low NC recovery may be associated to the poor selectivity of ultrasound waves acting on both, the amorphous and crystalline regions that may explain NC yields lower than 10% (w/w) [17].”
Q2. The quality of the images in figure 1 must be improved. Also, the figure must be depicted deeply and discussed.
Response
Comment acknowledged.
Figure 1 was replaced, showing better resolution, but this time is shown as Figure 2. In addition, section 2.2.4 was enriched with other micrographs to discuss particle size and agglomerates formation, which affected films topography. This is written in Lines 218-234 as follows:
“Figure 3 shows micrographs of NC and CH-NC films. Figure 3A shows the particle size of NC obtained by AFM where 15-20 particles were selected from different areas of the film, ranging between 74.63 and 128.85 nm.
According to Boukouvala et al. [30], cellulose may be considered as nanocrystal if the crystalline particles range from 1 to 1000 nm. The observed particles size is the result of the type of extraction applied, and some authors consider that size is proportional to the degree of polymerization. Hydrolysis carried out with sulfuric acid (64% v/v) allowed crystals formation [31].
Figure 3B shows the micrograph of a NC-CH film, for an area of 5 x 5 mm, in which the formation of agglomerates can be observed. According to Börjesson et al. [31], for films made with NC and dried by evaporation in contact with air, there is induction of NC agglomerates during film formation.”
Figure 2. Films topography. A) Cellulose-Chitosan film, B) Cellulose-Nanocrystals-Chitosan film, C) Chitosan film. Ra: Arithmetic roughness average of the surface; Rq: Root mean square average of the profile heights over the evaluation length. Different lowercase letters (a-b) next to reported values in the same parameter indicate significant difference (p<0.05).
Figure 3. A) Cellulose nanocrystals showing selected particles size range of 74.63-128.85 nm. B) Films of cellulose nanocrystals (CN) and chitosan (CH).
Q3. Line 193. Which is the depiction that belongs to A), B), and C)?
Response
Comment acknowledged.
We added the description of Figure 4 (FTIR spectrograms) on lines 224-226, as follows:
“The spectrograms of the films produced from corn cob cellulose (C), cellulose-chitosan (C-CH), and cellulose-cellulose nanocrystals-chitosan (C-NC-CH), are shown in figures 4A, 4B and 4C, respectively.”
Q4. Section 2.2.6. What was the rationale for monitoring the biodegradability only on day 0 and day 28? It could be interesting to evaluate the biodegradation of the films between days 0 and 28.
Response
Comment acknowledged.
Biodegradability was evaluated every 5 days, the images of this test are presented in Figure 6. According to the ASTM standard, the incubation time required is 28 d. We include this information on Lines 282-284 as follows:
“The biodegradability test was carried out for 28 days, since the G21-09 standard establishes 28 days of incubation as the standard duration of the test. Testing can be completed in less than 28 days for samples showing a growth index of two or more (10-30% of area covered).
We have included a Figure (Figure 5) which the biodegradability progress of films is presented. We include a description of this was added in line 318-322, as follows:
“The progress of biodegradability can be observed in Figure 6 at 5 d, 10 d, 15 d, 20 d, 25 d and 28 d after inoculation. According to the standard, testing can be completed in less than 28 d for samples showing a growth index of two or more (10-30% of area covered), and thus, the study could be stopped after 10 d for the cellulose film, and the control (filter paper). However, no growth was observed on the surface of the chitosan film after 28 d of analysis, while the NC film presented >10% of its surface covered by fungi at 20 d.”
Figure 5. Biodegradability progress of cellulose films (A), cellulose nanocrystals (B), chitosan (C) and filter paper (D) observed on d 5, 10, 15, 20, 25 and 28 after inoculation of spore mixture.
Q5. Revise the order of the section numbers.
Response.
Comment acknowledged.
The numbering was corrected
Q6. Section 4.2.2. How do authors evidence they extracted nanocrystals
Response.
Comment acknowledged.
We added two micrographs to explain how the NC particles size determined, and we also added a brief discussion of this in Lines 201-203 and 212-221, as follows:
Figure 3 shows micrographs of NC and CH-NC films. Figure 3A shows the particle size of NC obtained by AFM where 15-20 particles were selected from different areas of the film, ranging between 74.63 and 128.85 nm.
According to Boukouvala et al. [30], cellulose may be considered as nanocrystal if the crystalline particles range from 1 to 1000 nm. The observed particles size is the result of the type of extraction applied, and some authors consider that size is proportional to the degree of polymerization. Hydrolysis carried out with sulfuric acid (64% v/v) allowed crystals formation [31].
Figure 3B shows the micrograph of a NC-CH film, for an area of 5 x 5 mm, in which the formation of agglomerates can be observed. According to Börjesson et al. [31], for films made with NC and dried by evaporation in contact with air, there is induction of NC agglomerates during film formation.”
Figure 3. A) Cellulose nanocrystals showing selected particles size range of 74.63-128.85 nm. B) Films of cellulose nanocrystals and chitosan (NC-CH).
In addition, in the materials section 3.2.4, the methodology used for this determination was added, line 409-416 as follows.
"3.2.4. Nanocellulose crystals size
Crystals size were determined using an atomic force microscope (AFM: Park NX10, Seoul, Korea), applying the no contact method and using aluminum coated silicone tip PPP-FMR (Nanosensors, PointProbe, Neuchatel, Switzerland) with a resonance frequency of 286-362 kHz and a spring constant of 20-80 N m-1. Samples of 0.5 x 0.5 cm were ana-lyzed and three 5 x 5 μm areas were scanned at a speed of 1 Hz with a resolution of 256 x 256 pixels [47]. Particle size of 20 particles was determined for each area of 5 μm x 5 μm, the measurement was made of 5 different films and 3 different areas of each one."

Reviewer 2 Report
This manuscript presented an interesting study about the properties and biodegradability of films based on cellulose and nanocellulose. The work has some potential. However, some points listed below need to be improved.
Introduction: please clearer the novelty of this work.
After Introduction section I suggest add the Material e Methods section.
Please check the section numbers. There is not section 3 in the manuscript.
Section 2.1: what is the yield of cellulose and nanocelulose from the literature? Please improve the discussions in this section.
Section 2.2.2: I suggest add SEM images of the samples to verify the supposed “agglomerates formation” and “induced stress points” related by the authors to explain the film mechanical properties.
Section 2.2.3: if possible add images of the obtained films.
Section 4.2.5: the authors must add the number of specimens tested in each test.
Author Response
Q1. Introduction: please clearer the novelty of this work.
Response
Comment acknowledged.
We included the novelty in Line 42-53, and 58-63, as follows
“The development and application of biopolymer-based films from agricultural by-products or waste food has increased lately, due to concerns about the overexploitation of limited natural resources such as fossil fuels, and the high environmental impact of packaging made from non-biodegradable materials.
Biopolymeric films meet the general characteristics of quality and appearance for food products, but also for public health, which increases consumer interest [1]. Poly-saccharides have emerged as one of the main biodegradable polymers for film production due to their high abundance and low toxicity; additionally, they are generally easy to make and exhibit good mechanical properties [3].
Agro-industrial waste can be obtained from trees or plants such as sugar cane bagasse, corn, bamboo, among others. The main fibrous residues produced by agricultural activity are known as cellulose fibers, and they have a high potential to be used as reinforcement in the manufacture of packaging materials due to their abundance, low weight, biodegradability and low cost. Assuming that 40% of agricultural production comprises by-products, and that at least 10% of them can be obtained as fiber, millions of metric tons of fiber would be available every year [5].”
Q2. After Introduction section I suggest add the Material e Methods section.
Response
Comment acknowledged.
However, the journal's author guide indicates that the presentation of the topics should be in the following order: Introduction, Results, Discussion, Materials and Methods, and Conclusion. With the option to place Results and Discussion in the same section.
Q3. Please check the section numbers. There is not section 3 in the manuscript.
Response.
Comment acknowledged.
The numbering was corrected
Q4. Section 2.1: what is the yield of cellulose and nanocellulose from the literature? Please improve the discussions in this section.
Response
Comment acknowledged. We restructured the section 2.1 Line 104 -111 as follows:
“2.1. Yield of Cellulose and Cellulose nanocrystals
The yield of C obtained from corn cob in this work was lower (21.28%±1.31%, by weight) than those reported by other authors of about 30% [14]. The low yield may be attributed to material losses due to repeated washings. The yield of NCs from the corn cob using acid hydrolysis and ultrasound was 5.70±1.71 (w/w), which coincides with those reported by other authors that are close to 6% (w/w). The low NC recovery may be associated to the poor selectivity of ultrasound waves acting on both, the amorphous and crystalline regions that may explain NC yields lower than 10% (w/w) [17].”
Q5. Section 2.2.2: I suggest add SEM images of the samples to verify the supposed “agglomerates formation” and “induced stress points” related by the authors to explain the film mechanical properties.
Response
Comment acknowledged.
SEM analysis were not carried out. Nevertheless, applying AFM, micrographs of larger areas were obtained where CN agglomerates can be observed. The micrograph of the CN-CH film and a brief discussion of the formation of the agglomerates were added in section 2.2.4. line 218-221 as follows:
“Figure 3B shows the micrograph of a NC-CH film, for an area of 5 x 5 mm, in which the formation of agglomerates can be observed. According to Börjesson et al. [31], for films made with NC and dried by evaporation in contact with air, there is induction of NC agglomerates during film formation.”
Figure 3. A) Cellulose nanocrystals showing selected particles size range of 74.63-128.85 nm. B) Films of cellulose nanocrystals and chitosan (CN-CH).
Q6. Section 2.2.3: if possible, add images of the obtained films.
Response
Comment acknowledged. We have included one new Figure (Figure 1), which shows the appearance of the films.
Figure 1. Appearance of edible films. C: Cellulose; NC: Cellulose Nanocrystals; CH: Chitosan.
Q7. Section 4.2.5: the authors must add the number of specimens tested in each test.
Response
Comment acknowledged.
To obtain representative properties of the films, measurements were made on 5 different sections of the film. This was repeated on 3 different films, the results presented are the average of the 15 measurements ± the standard deviation.
This information has been described in section 3.2.6.11 line 552-554 as follows:
“To obtain representative results, five different sections of three different films were tested. The results are the average of these measurements ± standard deviation.”

Round 2
Reviewer 2 Report
After corrections the manuscript reads well. I suggest publication in its current form.